# Complex Modification Orders Alleviate the Gelling Weakening Behavior of High Microbial Transglutaminase (MTGase)-Catalyzed Fish Gelatin: Gelling and Structural Analysis

**DOI:** 10.3390/foods12163027

**Published:** 2023-08-11

**Authors:** Kaiyuan Su, Wanyi Sun, Zhang Li, Tao Huang, Qiaoming Lou, Shengnan Zhan

**Affiliations:** 1College of Food and Pharmaceutical Science, Ningbo University, Ningbo 315211, China; sukaiyuan327@163.com (K.S.); 18071965076@163.com (W.S.); 18757418127@163.com (Z.L.); louqiaoming@nbu.edu.cn (Q.L.); 2Zhejiang-Malaysia Joint Research Laboratory for Agricultural Product Processing and Nutrition, Ningbo University, Ningbo 315211, China

**Keywords:** gelatin gels, modification orders, gelling properties, modification mechanism

## Abstract

In this paper, the effects of different modification orders of microbial transglutaminase (MTGase) and contents of pectin (0.1–0.5%, *w*/*v*) on the gelling and structural properties of fish gelatin (FG) and the modification mechanism were studied. The results showed that the addition of pectin could overcome the phenomenon of high-MTGase-induced lower gelling strength of gelatin gels. At a low pectin content, the modification sequences had non-significant influence on the gelling properties of modified FG, but at a higher pectin content (0.5%, *w*/*v*), P_0.5%_-FG-TG had higher gel strength (751.99 ± 10.9 g) and hardness (14.91 ± 0.33 N) values than those of TG-FG-P_0.5%_ (687.67 ± 20.98 g, 12.18 ± 0.45 N). Rheology analysis showed that the addition of pectin normally improved the gelation points and melting points of FG. The structural results showed that the fluorescence intensity of FG was decreased with the increase in pectin concentration. Fourier transform infrared spectroscopy analysis indicated that the MTGase and pectin complex modifications could influence the secondary structure of FG, but the influenced mechanisms were different. FG was firstly modified by MTGase, and then pectin (P-FG-TG) had the higher gelling and stability properties.

## 1. Introduction

Gelatin is a class of biopolymers which is widely used in modern manufacturing industries, especially in food processing, pharmaceutics, and cosmetics, due to its unique functional properties (e.g., foaming, film forming, emulsification, biocompatibility, biodegradability) [1]. Gelatin is extracted from various sources, the primary sources of which are mammals (cows and pigs). But the demand for extracting it from alternative sources is increasing for religious–cultural–health reasons. Promising sources are aquatic (mostly fish) and poultry [2,3].

Compared with mammalian gelatin, fish gelatin (FG) has lower proline and hydroxyproline contents, and these amino acids are responsible for the stability of collagen like the triple helix structure of gelatin. Thus, FG has lower gel strength and gelling and melting temperatures, which limit its application [2]. In order to overcome these shortcomings, many modification methods have been proposed to modify FG, such as enzyme modification, chemical modification, mechanical treatment, and electrostatic interaction methods [4]. Among them, enzymatic modification and electrostatic interaction are effective methods to alter the molecular structure and functional properties of FG [5]. For enzyme modification, microbial transglutaminase (MTGase) can catalyze the covalent crosslinking of protein peptides within protein molecules, effectively changing the structure and function of proteins [6,7]. However, excessive MTGase may hinder inter-molecular aggregation, thereby reducing the formation of protein gel networks and finally producing thermally irreversible gel with low strength and hardness [8,9]. Our previous report showed that the addition of pectin can overcome this weakening phenomenon [10]. Our previous work has shown that pectin–MTGase-complex-modified FG had better rheological properties and higher gel strength and melting temperature than those of the original one, because this complex modification largely introduced lots of hydrogen bonds that decreased the excessive formation of covalent bonds. However, the effect of complex modifications order on the gelling and structure properties of FG has not been clarified [10,11,12].

Therefore, this study mainly investigates the orders of MTGase and pectin complex modifications on the gelling and rheological properties of FG through gel strength, texture profile analysis (TPA), and rheological analysis. Moreover, the modification mechanisms were also discussed through the analysis of endogenous fluorescence, FTIR, and gels structure. We hoped this paper would mainly provide an important theoretical basis for the application of complex modifications for the improvement of the functional properties of gelatin.

## 2. Materials and Methods

### 2.1. Material

Tilapia skin gelatin (type A, crude protein 88.14 ± 1.57%, ash 4.78 ± 0.03%, moisture 7.93 ± 0.63%, dry basis) was purchased from the Yuanye Company (Shanghai, China). Microbial transglutaminase (MTGase, 100 U/g) was obtained from Yiming Company (Taixing, China). High-methoxyl citrus pectin (galacturonic acid content, 79%; degree of methyl esterification, 71%) was obtained from Aladdin (Shanghai, China). All other chemical reagents are analytical grade.

### 2.2. Preparation of Gelatin Gels

Fish gelatin (FG) solution (6.67% *w*/*v*, pH 6.5) was prepared through dissolving FG in non-ionic water at 40 °C. Then, the orders of modification FG by MTGase (0.005%), various contents of pectin (0.1%, 0.3%, 0.5%) were divided into two groups: TG-FG-P and P-FG-TG. TG-FG-P: Firstly, different concentrations of pectin were added to FG solution and stirred at 40 °C until the pectin was completely dissolved. Then, MTGase was added into the mixture solution and cross-linked at 40 °C for 40 min. The enzyme was inactivated at 100 °C for 5 min. P-FG-TG: MTGase was firstly used to modify FG, and the mixture was incubated at 40 °C for 40 min for the cross-linking. And then, the enzyme was inactivated at 100 °C for 5 min. Subsequently, different concentrations of pectin were added into the mixture and stirred at 40 °C until completely dissolved to form stable composite colloid. After reaction, all the gelatin solution was finally cooled down to form a stable composite colloid and stored at 4 °C for further use.

### 2.3. Gelling Properties

#### 2.3.1. Gel Strength

According to the method of Sha et al. [13] with minor modification, 15 mL gelatin solution (6.67%, *w*/*v*) was transferred into a 25 mL glass beaker, and then put into a refrigerator at 4 °C for 16–18 h to form the gels. The gel strength was measured using a texture analyzer (Stable Micro System, Surrey, UK) with a probe of P 0.5R (12.7 mm diameter flat-faced cylindrical plunger). The gel strength was recorded at a speed of 1.00 mm/s when the probe penetrated the gel for 4 mm. Each sample was performed in triplicates.

#### 2.3.2. Texture Profile Analysis (TPA)

TPA of gelatin gels was performed using a TA.XT plus texture analyzer (Stable Micro System, Surrey, UK) equipped with the Peltier temperature control system (4 °C). Gelatin solution (6.67%, *w*/*v*) was put into a cylindrical container and matured in a refrigerator at 4 °C for 16–18 h. The gels were subjected to two cycle compressions to 40% of their original height at a speed of 1 mm/s with a P 36R probe (47 mm diameter flat cylindrical probe). The deformation of gel was 40%, and each sample was tested three times. Finally, the hardness, elasticity, cohesion, gelation, and chewiness of the colloid were obtained [14].

### 2.4. Gelation Point and Melting Point Analysis

The storage modulus (G′) and loss modulus (G″) of all gelatin solution (6.67%, *w*/*v*) were measured using a controlled rheometer (Discovery Hybrid Rheome-ter, TA Instrument, Surrey, UK) with a stainless-steel parallel plate (60 mm diameter). Silicone oil was smeared evenly around the parallel plate before experiment. The gap between the fixture and base was 1000 μm. The FG solution was first cooled from 40 °C to 4 °C, kept at 4 °C for 5 min, and then heated from 4 °C to 40 °C, with a rate of 0.5 °C/min, a strain force of 0.5%, and at a frequency of 1 Hz. The gelation point (*Gp*) and melting point (*Mp*) could be obtained during the cooling and heating procedure, respectively [10,15].

### 2.5. Transparency

The transparency of all gelatin solutions was determined with an ultraviolet visible spectrophotometer at a wavelength of 600 nm [8,16]. Distilled water was used as the blank control, each sample was measured three times, and the calculated result was expressed as transparency: y = −$\begin{array}{l} \log Abs600\\ \;x \end{array}$, where Abs_600_ is the absorbance at 600 nm, and *x* is the cup thickness (mm).

### 2.6. Structure Properties

#### 2.6.1. Fluorescence Intensity

A fluorescence spectrophotometer (F-47000, Hitachi, Ibaraki, Japan) was used to measure the fluorescence intensity for all gelatin solution [17]. The 2 mg/mL gelatin solution was transferred to the cuvette, and the excitation wavelength was set at 280 nm. In the range of 280 nm to 460 nm, the scanning speed was 100 nm/min at room temperature. The excitation and emission slit widths were both set at 2.5 nm.

#### 2.6.2. Fourier Transform Infrared Spectroscopy (FTIR)

The FTIR analysis of samples was performed using an FTIR spectrometer (Perkin Elmer, Waltham, MA) [18]. The lyophilized sample was mixed with KBr powder evenly and then pressed into a tablet. The spectra were collected in the infrared range of 4000–500 cm^−1^ at 25 °C. The obtained automatic signals were collected in 32 scans with a resolution of 4 cm^−1^, and each sample was subjected to three parallel tests.

#### 2.6.3. Scanning Electron Microscope

According to the method of María S et al. [19], the micro-structure of the composite gels was observed using an environmental scanning electron microscope (Fei Deutschland GmbH, Dreieich, Germany). The lyophilized sample was cut into small pieces (about 2 × 2 × 2 mm), and then placed on the conductive adhesive and coated with gold. The micro-structure of all gelatin gels was tested with 10 kV accelerating voltage. All microscope pieces were 1000×.

## 3. Results and Discussion

### 3.1. Gel Strength and TPA

Gel strength is one of the main physical properties of gelatin, and it is a stiffness factor that can predict the quality of gelatin. TPA can better simulate the effect of the tongue and teeth on gel [8]. As shown in Table 1, FG-TG has lower gel strength and hardness values than those of control. This might be because the high contents of MTGase can catalyze FG to form excessive inter–intra covalent bonds, weakening the uniform structure [9], while in the TG-FG-P group, with the increase in pectin concentration, the gel strength and hardness of FG first increased (0.1%) and then decreased (0.3–0.5%); TG-FG-P_0.1%_ gel displays the highest gel strength and hardness values. The addition of a certain amount of pectin can form a stable FG-P polymer mixture through electrostatic interaction [7]. Following this, MTGase catalyzed the FG-P polymer mixture to form a stronger network structure via covalent cross-linking among complexes or FG. However, for the high pectin contents, phase separation might occur that disturbs the stability of the FG-P polymer mixture. MTGase cannot catalyze the FG-P polymer mixture to form a stable mixed system, resulting in a downward trend of gel strength and hardness [10]. For the P-FG-TG group, the gel strength and hardness of gelatin gels are increased with the increase in pectin content, and P_0.5%_-FG-TG has the highest values. MTGase firstly catalyzes lysine and glutamine in FG to form super polymers, and the addition of pectin still could interact with super-FG molecules through electrostatic interactions to form macro-molecular polymers, enhancing the hydrogen bonds of the gelatin gels and increasing the gel strength and hardness values of FG-TG composite colloids with the increase in pectin concentration. The gumminess and chewiness of all composite colloids almost show a similar trend of gel strength and hardness. Interestingly, the three modification methods have non-significant differences on the springiness, cohesiveness, and resilience of all the gelatin gels. Similarity, Huang et al. [8,10] also reported that MTGase and pectin complex modification, MTGase modification, and pectin modification could not change the springiness, cohesiveness, and resilience of gelatin gels. In conclusion, the tendency of the gel strength and hardness values of TG-FG-P and P-FG-TG are different, and the P-FG-TG group could alleviate the gelling weakening behavior of high-MTGase-catalyzed FG.

### 3.2. Gelation Point and Melting Point Analysis

The dynamic viscoelastic behavior (DVB) of gelatin samples in the temperature range of 40~4 °C and 4~40 °C was monitored to obtain the gelling point (*Gp*) and melting point (*Mp*) [20]. As shown in Figure 1a,b, the loss modulus (G″) is always greater than the storage modulus (G′) at the initial stage. With the decrease in temperature, the G′ and G″ of all gelatin samples increased significantly, and G′ exceeds G″, suggesting that triple-helix structures in gelatin samples are forming [10,21]. At 4 °C, the G′ of the two groups is equivalent, which indicates that the elasticity of the two groups of gelatin is basically the same at this time. The G′ of the TG-FG-P group is lower than that of the P-FG-TG group, which indicates that the viscoelasticity of the P-FG-TG group gelatin solution is better when the gelatin sample is semi-solid. At the same time, the G″ of P_01%_-FG-TG is also higher than that of P_0.5%_-FG-TG. This shows that the molecular arrangement of FG modified by MTGase and then added with low-concentration pectin is denser with better viscoelasticity and higher molecular stability than other samples. Figure 1c,d shows that during the heating process (4–40 °C), the G′ and G″ of gelatin began to decline from the initial low temperature with high modulus state to solution, which also shows that the composite FG samples are still thermally reversible, and the gelatin can recover to the original molecular state [22].

The crossover points of G′ and G″ on the cooling and heating curves are defined as the *Gp* and *Mp* [15,23]. It can be seen from Table 2 that with the increase in pectin concentration, the *Gp* and *Mp* of the gelatin samples slightly increased, and all the complex modified gelatin gels had higher *Gp* and *Mp* values than those of pure FG. However, the modification orders of MTGase and pectin have non-significant influence on the *Gp* and *Mp*. For the TG-FG-P group, FG interacts with pectin to form a super-polymer; MTGase catalyzes this super-polymer or cross-links with gelatin to form an interconnected molecular ring structure, thus improving the *Gp* and *Mp*. For the P-FG-TG group, MTGase catalyzed FG molecules to form double bonds; after the deactivation of MTGase, pectin and FG were polymerized through hydrogen bonding to form super-polymers, thus improving the *Gp* and *Mp* of gelatin. Nevertheless, the TG-FG-P group and P-FG-TG group had higher *Gp* and *Mp* values than those of FG-TG, especially the complexes with high pectin contents. This suggested that the MTGase and pectin complex modification had a higher improvement efficiency than that of pure MTGase modification.

### 3.3. Turbidity Analysis

Turbidity represents the formation of insoluble compounds. The size of turbidity is not only related to the content of colloidal substances in water, but also to the size and shape of these substances, usually closely related to the stability of protein polysaccharide systems [24]. The higher the turbidity, the lower the transparency. As shown in Table 1, the turbidity values of the mixed colloidal solution in the two groups are greater than FG, and it is increased with the increase in pectin concentration, which indicates that there is strong electrostatic attraction between FG and pectin in the system. The FG-P system self-assembles to form large insoluble polymer molecules at the same time, and the related phases of the molecules are separated, while the accumulation of a large number of insoluble compounds (aggregates) leads to the highest turbidity of the system [25]. With the increase in pectin concentration, the pH value of the colloid also decreases, which makes it easier to form polymers. The turbidity in the two groups of gelatin samples increased significantly and the difference was also very small. Thus, the modification decreased the transparency of gelatin, but modification orders had non-significant influence on transparency.

### 3.4. Structure Properties

#### 3.4.1. Fluorescence Analysis

Tyrosine and phenylalanine in gelatin are the main fluorescent substances, and fluorescence analysis can reflect the minor changes in the tertiary structure of protein [26,27]. According to Figure 2, the peak of FG-TG at 303 nm is higher than that of the original FG, and the fluorescence intensity significantly increases. This may be because MTGase catalyzes gelatin cross-linking, thereby changing the protein structure and promoting the exposure of fluorescent groups (tyrosine, phenylalanine). The fluorescence intensity of the TG-FG-P group (Figure 2a) decreased except for TG-FG-P_0.3%_. Studies have shown that pectin has a quenching effect on the fluorescence groups of gelatin [28]. When FG is decorated with a small amount of pectin, pectin combines with the carboxyl groups in phenylalanine and tyrosine to form a small amount of complex, quenching the fluorescence groups of gelatin. Excessive pectin will quench most of the fluorescent substances in FG, leading to a significant decrease in the fluorescence intensity of the gelatin solution. For the P-FG-TG group (Figure 2b), the fluorescence intensity of the gelatin solution is decreased with the increase in pectin concentration. Though MTGase firstly cross-links with FG to expose the fluorescence sites of FG, pectin still can interact with gelatin; that quenches the fluorescence groups in FG, and more fluorescence groups will be quenched with the increase in pectin. Interestingly, the overall fluorescence intensity of the P-FG-TG group was higher than that of the TG-FG-P group. This may be because pectin firstly quenched many of the fluorescent amino acids in FG. MTGase cannot induce the formed FG-P complex exposure of fluorescent groups. In conclusion, the fluorescence intensity of gelatin will be reduced to varying degrees depending on the modification orders of pectin and MTGase.

#### 3.4.2. FTIR Analysis

FTIR can be used to analyze the changes in functional group information attached to protein skeletons, reflecting changes in the secondary structure of proteins [29]. Amide A represents the stretching vibration of—NH_2_ residues, which is related to the formation of intra-molecular hydrogen bonds. As shown in Figure 3, amide A of FG-TG is located at 3311.18, which is lower than that of FG (3316.00). This is because MTGase catalyzes FG to form covalent bonds, reducing the hydrogen bonds of—NH_2_, leading to the reduction of amide A. Normally, the amide A of P-FG-TG is lower than that of TG-FG-P. This because in P-FG-TG, MTGase firstly catalyzes the cross-linking of FG to form covalent bonds, reducing the amino acids that can form hydrogen bonds, resulting in fewer hydrogen bonds formed when pectin interacts with FG. In the TG-FG-P group, pectin interacts with FG to form a stable complex via the formation of hydrogen bonds. At this time, fewer catalysis sites could be used to form covalent bonds among FG molecules with the influence of MTGase. Thus, the TG-FG-P group had less covalent bonding. This might explain why P-FG-TG had higher gel strength and hardness than those of the TG-FG-P group. Amide I mainly exhibits C=O stretching, while amide II mainly exhibits N-H deformation [30]. The amide I of both FG and FG-TG is 1647.88, while the addition of pectin generally increased the amide I values, indicating that the catalytic effect of MTGase on FG does not affect the C=O groups in gelatin. But, the addition of pectin was attributed to the formation of N-C bonds between the NH_2_ of gelatin and C=O of pectin, which was due to the increase in inter-molecular hydrogen bonds. Thus, both complex modification orders could improve the gelling properties of TG-modified FG.

#### 3.4.3. Electron Microscope Scanning

In order to better observe the modification orders on the microstructure of modified gelatin gels, SEM was used to analyze the sub-micron gel network structure [10]. As shown in Figure 4, the pure FG gels have larger holes, but modified gelatin gels have a much denser gel network with fewer and smaller-size holes. For the TG-FG-P group, the sizes of the pores are decreased with the increase in pectin concentration (0.1–0.3%), and then increased. TG-FG-P_0.3%_ has the densest network with smaller numbers of pores. Low-concentration pectin can be trapped in the original pores through the formation of inter-molecular hydrogen bonds, reducing the catalytic sites of MTGase, forming FG-P and FG-FG network structures [7,10]. However, as a typical weak gel, excessive pectin (0.5%) will lead to instability of the FG–pectin complex, thus breaking the stable structure of gels. This shows that that pectin and FG molecules are connected to form a network structure through hydrogen bonds and inter-molecular forces, and MTGase can make the molecules in the composite colloid form inter-molecular double bonds for cross-linking, so as to achieve a more dense network structure [31], improving the gelling properties (Table 1), Gp, and Mp (Table 2) of FG. For the P-FG-TG group, with the increase in pectin concentration, the gelatin gels gradually formed a special state of a large ring wrapping a small ring. MTGase firstly catalyzed FG to form super-polymers with fewer pores; with the increase in pectin concentration, super-FG molecules are wrapped in the macro-molecular ring of pectin condensation, and the molecules inside the ring are connected by the double bonds, thus forming the stable and bigger ring with a small ring. Thus, the modified gelatin gels also had higher gelling properties (Table 1), Gp, and Mp (Table 2) than those of pure FG and FG-TG.

In conclusion, the gel network of complex modified gelatin is the most uniform and fine, which indicates that the complex modification can overcome the phenomenon “excessive MTGase weaken gelling properties of FGG″.

### 3.5. Graphical Representation of the Gelation Mode

In order to study the effects of different modification orders of MTGase and pectin on the gelling properties and molecular structure of FG, the modification mechanisms were established. As shown in Figure 5, for pure FG gelation, FG molecules are orderly arranged and connected to form a colloidal network structure through inter- and intra-molecular forces. MTGase catalyzes FG molecules to form covalent bonds. In the TG-FG-P group, pectin and FG are firstly interconnected through inter-molecular hydrogen bonds to form a macro-molecular polymer. MTGase catalyzed the free FG or exposed crosslink sites of FG-P coacervates to form covalent bonds, thereby forming a relatively denser and more stable gel structure. And the addition of pectin could overcome the results of “high MTGase weaken gelling”. However, a higher concentration of pectin will mask the crosslink sites of the FG site or cause phase separation, thus reducing the gelling properties. In the P-FG-TG group, MTGase firstly catalyzed FG to form super gelatin molecules with inter- or intra-covalent bonds, and then the addition of pectin still can interact with super or free FG molecules to form stable super-FG-P coacervates via the formation of hydrogen bonds with the increase in pectin content, and pectin surrounds the FG. Thus, compared with pectin-MTGase modification (TG-FG-P), MTGase-pectin modification (P-FG-TG) might be much more proper for producing gelatin with higher stability and gelling properties.

## 4. Conclusions

Herein, the different complex modification orders of MTGase and pectin on the gelling and structural properties of FG were comparatively investigated. The results showed that two kinds of complex modification orders could overcome the drawback of high MTGase causing gelling weakness, and almost all complex modified gelatin gels had higher gelling properties (gel strength, hardness, *Gp* and *Mp* values) than those of pure FG and FG-TG. At the same pectin content, compared with the TG-FG-P group, the P-FG-TG group had slightly higher gelling properties and stability, and P_0.5%_-FG-TG had the highest gelling properties. Structural results showed that complex modified gelatin gels had the densest gel network compared to both FG and TG-FG, but the TG-FG-P group had fewer covalent bonds than the P-FG-TG group. Through the viewpoints of gelling properties, MTGase-pectin modification (P-FG-TG) could be used to produce gelatin with high quality. However, further studies still should be performed to evaluate its true application value in the food, e.g., yoghurt, and release delivery systems.

## Figures and Tables

**Figure 1 foods-12-03027-f001:**
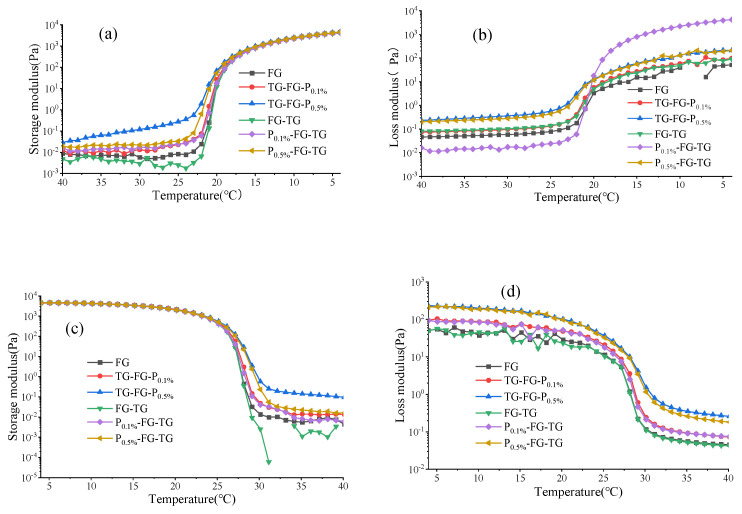
Effect of different modification sequences of MTGase and pectin on the G′ and G″ of gelatin during the cooling procedure (**a**,**b**) and the heating procedure (**c**,**d**).

**Figure 2 foods-12-03027-f002:**
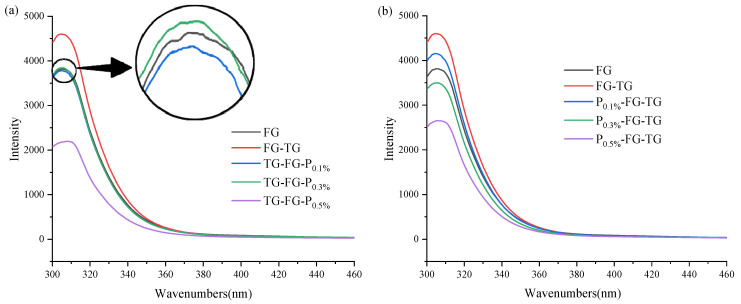
Fluorescence spectra of TG-FG-P (**a**) and P-FG-TG (**b**).

**Figure 3 foods-12-03027-f003:**
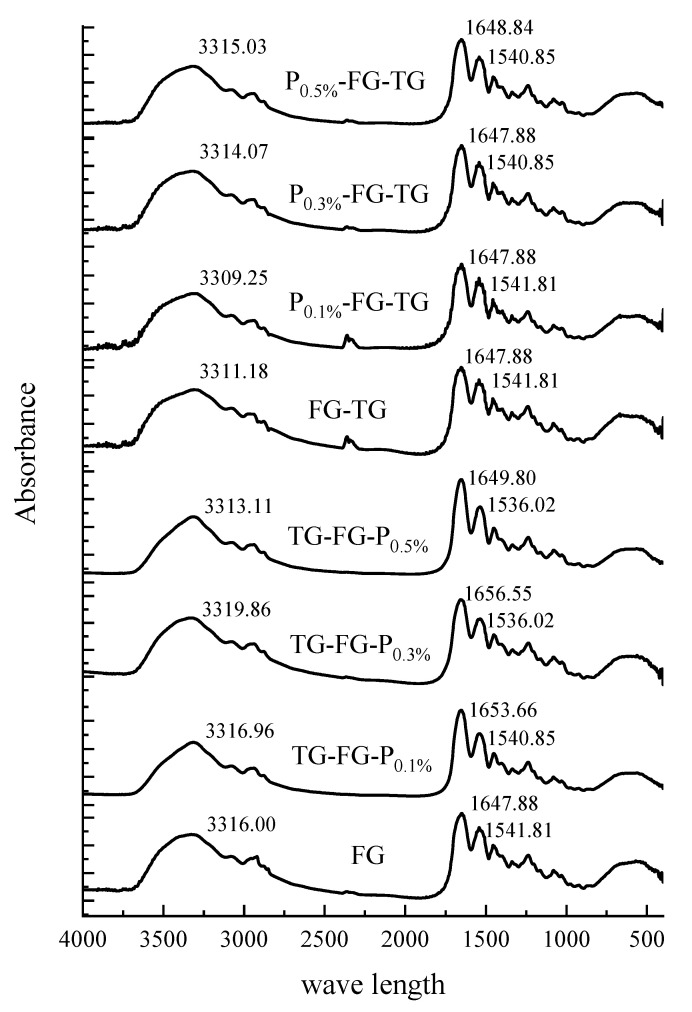
FTIR spectra of different-modification-order modified gelatin.

**Figure 4 foods-12-03027-f004:**
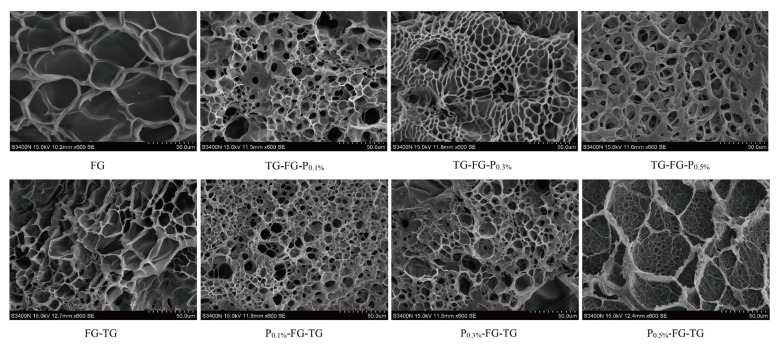
Effect of different modification orders of TG and pectin on the microstructure of gelatin gels.

**Figure 5 foods-12-03027-f005:**
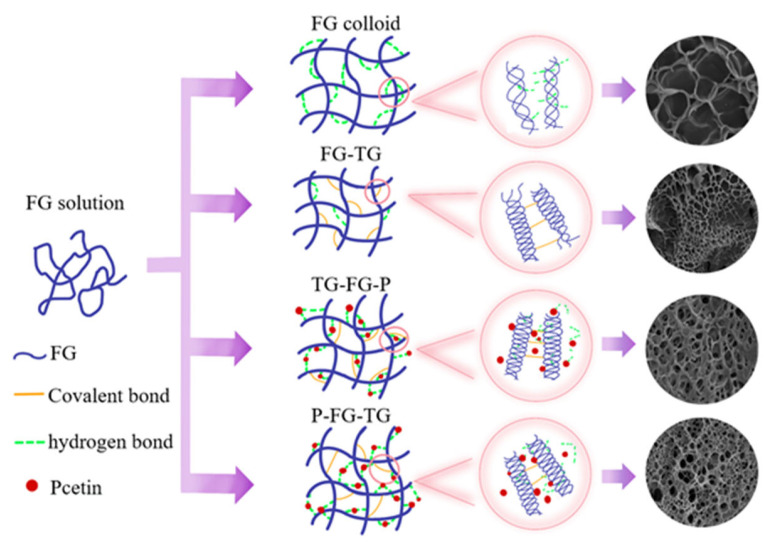
Structural diagram of the effect of different modification orders of TG and pectin on an FG molecule.

**Table 1 foods-12-03027-t001:** Effect of different modification orders of MTGase and pectin on the gelling and stability properties of FG.

Content	TG-FG-P				P-FG-TG			
Parameters	FG	TG-FG-P_0.1%_	TG-FG-P_0.3%_	TG-FG-P_0.5%_	FG-TG	P_0.1%_-FG-TG	P_0.3%_-FG-TG	P_0.5%_-FG-TG
Gel strength(g)	701.78 ± 11.81 ^bc^	744.02 ± 5.45 ^a^	730.18 ± 11.43 ^ab^	687.67 ± 20.98 ^c^	685.84 ± 11.69 ^c^	728.35 ± 10.93 ^ab^	737.67 ± 12.05 ^ab^	751.99 ± 10.97 ^a^
Hardness(N)	13.51 ± 0.28 ^cd^	14.36 ± 0.27 ^ab^	13.98 ± 0.24 ^bc^	12.18 ± 0.45 ^de^	13.29 ± 0.28 ^cd^	14.46 ± 0.12 ^ab^	14.83 ± 0.28 ^a^	14.91 ± 0.33 ^a^
Springiness	0.95 ± 0.03 ^a^	0.96 ± 0.02 ^a^	0.95 ± 0.02 ^a^	0.97 ± 0.01 ^a^	0.97 ± 0.01 ^a^	0.95 ± 0.01 ^a^	0.97 ± 0.00 ^a^	0.96 ± 0.06 ^a^
Cohesiveness	0.92 ± 0.02 ^a^	0.93 ± 0.01 ^a^	0.92 ± 0.01 ^a^	0.92 ± 0.01 ^a^	0.93 ± 0.01 ^a^	0.92 ± 0.02 ^a^	0.93 ± 0.01 ^a^	0.91 ± 0.01 ^a^
Gumminess(N)	12.65 ± 0.58 ^b^	13.36 ± 0.23 ^ab^	12.92 ± 0.31 ^ab^	11.25 ± 0.29 ^c^	12.31 ± 0.17 ^b^	13.37 ± 0.24 ^ab^	13.75 ± 0.20 ^a^	13.84 ± 0.30 ^ab^
Chewiness(N)	12.27 ± 0.27 ^ab^	12.81 ± 0.47 ^ab^	12.34 ± 0.54 ^ab^	10.96 ± 0.24 ^c^	11.99 ± 0.91 ^b^	12.75 ± 0.30 ^ab^	13.34 ± 0.19 ^ab^	13.65 ± 0.33 ^a^
Resilience	0.78 ± 0.01 ^b^	0.79 ± 0.01 ^ab^	0.79 ± 0.01 ^ab^	0.79 ± 0.04 ^ab^	0.79 ± 0.01 ^ab^	0.79 ± 0.03 ^ab^	0.80 ± 0.00 ^ab^	0.79 ± 0.00 ^ab^
Transparency	2.90 ± 0.00 ^a^	1.43 ± 0.00 ^b^	1.09 ± 0.00 ^c^	1.04 ± 0.00 ^c^	2.96 ± 0.00 ^a^	1.44 ± 0.00 ^b^	1.10 ± 0.00 ^c^	1.01 ± 0.00 ^c^

TG-FG-P: FG was firstly modified by pectin, and then MTGase modified the FG-P complex. P-FG-TG: FG was firstly modified by MTGase, and then pectin modified the FG-TG complex. The mean values in the same column with different lowercase letters (a–e) differ significantly (*p* < 0.05).

**Table 2 foods-12-03027-t002:** Effect of different modification orders of TMTGase and pectin on the *Mp* and *Gp* of gelatin gels.

Content	*Mp* (°C)	*Gp* (°C)
FG	27.41 ± 0.00 ^c^	20.56 ± 0.00 ^c^
TG-FG-P_0.1%_	27.99 ± 0.01 ^b^	20.89 ± 0.01 ^b^
TG-FG-P_0.5%_	28.85 ± 0.00 ^a^	21.61 ± 0.00 ^a^
FG-TG	27.79 ± 0.02 ^bc^	20.65 ± 0.01 ^c^
P_0.1%_-FG-TG	27.84 ± 0.00 ^bc^	20.71 ± 0.00 ^bc^
P_0.5%_-FG-TG	28.85 ± 0.01 ^a^	21.35 ± 0.00 ^a^

Different letters (a–c) in the same line indicate the significant differences (*p* < 0.05).

## Data Availability

The data presented in this article are available on reasonable request from the corresponding author.

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
