# Peer review of "Complex Modification Orders Alleviate the Gelling Weakening Behavior of High Microbial Transglutaminase (MTGase)-Catalyzed Fish Gelatin: Gelling and Structural Analysis"

_foods, 2023, doi:10.3390/foods12163027_

Round 1
Reviewer 1 Report (New Reviewer)
Dear Authors
I read your article carefully. I think the topic of study is important. The experiments performed are relatively sufficient. But in this situation, the following are suggestions for improving your manuscript. I hope you correct them well.
1- Title and keywords:
- The abbreviation MTGase is given for the first time in the manuscript without prior introduction. No abbreviations should appear in the entire text without a prior introduction in the manuscript.
- Accurate and correct keyword selection is one of the most important ways to improve and expand the searchability of the article after publication. The words used in the title and keywords are the same. Authors should use other keywords as manuscript keywords.
2- Abstract:
- Line 13, is MTGase an abbreviation of the words "Microbial transglutaminase" or of the words "micro glutamine transaminase" (what you put in the text)? I think it is an abbreviation of the words "Microbial transglutaminase". please check.
- Although this section is well written, I recommend you mention the study's limitations and their solutions briefly in one or two sentences.
3- Introduction: Gelatin is extracted from various sources, the primary sources of which are mammals (cows and pigs). But the demand for extracting it from alternative sources is increasing due to religious-cultural-health reasons. Promising sources are aquatic (mostly fish) and poultry. Please refer to these items in the text and use the following references in the text:
- https://doi.org/10.3390/foods12030670
- https://doi.org/10.3389/fsufs.2023.1172522
3- All references older than 2016 should be removed from the text and replaced with new references.
4- Results section, discussion, and figures and tables:
- Results and discussion are acceptable.
- Figures are acceptable.
- Is there a significant difference between the values of the average melting points of 27.41 and 27.99 at the 5% level? Please make sure.
5- Conclusion section:
- Revise and rewrite this section. In my opinion, the correct principles of writing research conclusions have not been fully observed and words like "we" have been used in the text.
with the best wishes
I think there is a few English mistakes in the text. Please check entire of manuscript carefully before publish.
Author Response
Comments and Suggestions for Authors
Dear Authors
I read your article carefully. I think the topic of study is important. The experiments performed are relatively sufficient. But in this situation, the following are suggestions for improving your manuscript. I hope you correct them well.
Response: Thanks for your valuable suggestions; we have revised the paper carefully according to your suggestions. Thanks very much.
1- Title and keywords:
- The abbreviation MTGase is given for the first time in the manuscript without prior introduction. No abbreviations should appear in the entire text without a prior introduction in the manuscript.
Response: Corrected, thanks.
- Accurate and correct keyword selection is one of the most important ways to improve and expand the searchability of the article after publication. The words used in the title and keywords are the same. Authors should use other keywords as manuscript keywords.
Response: We had used “Modification mechanism” instead of “Structural characteristics”. Other keywords are too representativeness of paper. Thanks.
2- Abstract:
- Line 13, is MTGase an abbreviation of the words "Microbial transglutaminase" or of the words "micro glutamine transaminase" (what you put in the text)? I think it is an abbreviation of the words "Microbial transglutaminase". please check.
Response: Yes, we had corrected it as "Microbial transglutaminase" throughout paper. Thanks.
- Although this section is well written, I recommend you mention the study's limitations and their solutions briefly in one or two sentences.
Response: Added. “Moreover, further studies should be performed to evaluate its true application value in the food processing, eg yoghurt, release delivery systems.”
3- Introduction: Gelatin is extracted from various sources, the primary sources of which are mammals (cows and pigs). But the demand for extracting it from alternative sources is increasing due to religious-cultural-health reasons. Promising sources are aquatic (mostly fish) and poultry. Please refer to these items in the text and use the following references in the text:
- https://doi.org/10.3390/foods12030670
Response: Added it as [1].
- https://doi.org/10.3389/fsufs.2023.1172522
Response: The title of this paper “Stress fusion evaluation modeling and verification based on non-invasive blood glucose biosensors for live fish waterless transportation”. We find the research content is not closely related to gelatin. We didn’t use it.Thanks.
3- All references older than 2016 should be removed from the text and replaced with new references.
Response: We had updated some older references, eg [6], [9], [16], [19], [23], [25], [26] and [28]. Thanks.
4- Results section, discussion, and figures and tables:
- Results and discussion are acceptable.
- Figures are acceptable.
- Is there a significant difference between the values of the average melting points of 27.41 and 27.99 at the 5% level? Please make sure.
Response: We had reanalyzed the data, and corrected them. Thanks.
5- Conclusion section:
- Revise and rewrite this section. In my opinion, the correct principles of writing research conclusions have not been fully observed and words like "we" have been used in the text.
Response: We had revised the conclusion, thanks very much.
Comments on the Quality of English Language
I think there is a few English mistakes in the text. Please check entire of manuscript carefully before publish.
Response: We had checked the paper throughout, and corrected some mistakes. Thanks very much.

Reviewer 2 Report (New Reviewer)
Lines 69-70. Erase “This group was named as TG-FG-P”.
Line 73. Erase “This group was named as P-FG-TG.”
The sample spent significant time at moderate and at high temperatures. How do the authors ensure the concentration of the samples? Under these conditions, the sample lost water. The authors should present the final solids content of the sample!
2.3.1 Gel strength Was only the gelling strength of the gelatin solution measured? What gelling concentration was used? What measurement temperature was used? How was the measurement temperature ensured?
The authors should describe the characteristics of the measuring accessory. what is a probe of P 0.5R?
2.3.2 Texture profile. Similar to the previous comment, the authors should describe the characteristics of the measurement fixture more appropriately, to which specimens the measurement was applied, and how they ensured that the measurements were carried out under constant 4 °C conditions. And how did they ensure that the measurements were carried out at constant conditions of 4 °C? What criteria did they employ to use a strain of 40%?
When the sample is heated or cooled in the rheometer, the gap between the geometry and the plate changes. How was this shrinkage or expansion compensated for?
The authors should improve the description of the sample to be measured. Only a 6.67% gelatin sample was measured, not the different mixtures. TG-FG-P and P-FG-TG.
Lines 103-105. Change transparency to turbidity.
The changes in TPA parameters are too small to be caused by structural changes in the materials. They could have been influenced by changes in chemical composition (specifically moisture content). As mentioned, the authors should have ensured the chemical composition of each sample.
Lines 138-143. The authors describe the formation of a coacervate phase. Did a coacervate phase form? They should present images of their produced hydrogels. Although the interaction between pectins and gelatins produces electrostatic complexes only under specific conditions, a coacervate phase is formed.
Fig 1. The authors present several curves where adequate analysis cannot be performed. It is recommended to add a graph of Tan delta as a function of temperature in order to have more tangible data for the reader.
An increase of less than one degree Celsius in the gelling temperature is not important for hydrogel applications. How many measurements were performed? It is crucial that there were at least three. It is also necessary to add the standard deviation of each measurement. Based on the above, it is required that the gelling point presents its standard deviation.
Author Response
Comments and Suggestions for Authors
Lines 69-70. Erase “This group was named as TG-FG-P”.
Response: Deleted.
Line 73. Erase “This group was named as P-FG-TG.”
Response: Deleted.
The sample spent significant time at moderate and at high temperatures. How do the authors ensure the concentration of the samples? Under these conditions, the sample lost water. The authors should present the final solids content of the sample!
Response: The aluminum foil was used to encase the cup during the samples preparation. We admitted that there was some water might flee through the steam evaporation. However, under the same condition, the samples were prepared. The less lost water had non-significant influence on the results. Moreover, herein, we used various contents of pectin, thus, the solids contents of samples were different. But, for the following experiment, the contents of gelatin was same, they were all 6.67% (w/v).
2.3.1 Gel strength. Was only the gelling strength of the gelatin solution measured? What gelling concentration was used? What measurement temperature was used? How was the measurement temperature ensured? The authors should describe the characteristics of the measuring accessory. What is a probe of P 0.5R?
Response: For this experiment, the gelatin content was still 6.67% (w/v). All the gelatin solution was poured into the beakers and held at 4 oC for the formation of gels. When the sample was fetched out form the refrigerator, the measurement was performed immediately. The test was very quick, which was less than 6 s. The P 0.5R is a common probe for the gel strength test. We had added some information to state it much clear. Thanks.
2.3.2 Texture profile. Similar to the previous comment, the authors should describe the characteristics of the measurement fixture more appropriately, to which specimens the measurement was applied, and how they ensured that the measurements were carried out under constant 4 °C conditions. And how did they ensure that the measurements were carried out at constant conditions of 4 °C? What criteria did they employ to use a strain of 40%?
Response: For this experiment, the gelatin content was still 6.67% (w/v). The gels were quickly performed after they were removed out from the cylindrical container. The test time was less than 10 s. Each sample test was performed one by one, and all tests were performed at room temperature. Moreover, before the experiment, we set the deformation was 40% in the measurement procedure. We had added some sentences to make it much clear. Thanks.
When the sample is heated or cooled in the rheometer, the gap between the geometry and the plate changes. How was this shrinkage or expansion compensated for?
Response: The constant gap value between the geometry and the plate was fixed as 1000 μm. We had added some sentences to make it much clear. Thanks.
The authors should improve the description of the sample to be measured. Only a 6.67% gelatin sample was measured, not the different mixtures. TG-FG-P and P-FG-TG.
Response: According to the results of gel strength and TPA, we mainly investigated the melting points and gelling points of FG, FG-TG, TG-FG-P0.1%, TG-FG-P0.5%, P0.1%-FG-TG and P0.5%-FG-TG. Besides, “gelatin solution” is a joint name of the samples.
Lines 103-105. Change transparency to turbidity.
Response: We had corrected them as “transparency” according to the reference. Thanks.
Fang, Q,; Ma, N,; Ding, K.Y,; Zhan, S.N,; Lou, Q.M,; Huang, T. Interaction between negatively charged fish gelatin and cyclodextrin in aqueous solution: Characteristics and formation mechanism, Gels. 2021,7(4), 260.
The changes in TPA parameters are too small to be caused by structural changes in the materials. They could have been influenced by changes in chemical composition (specifically moisture content). As mentioned, the authors should have ensured the chemical composition of each sample.
Response: For the sample preparation, we added the same volume of water. The variable factors were pectin contents (0, 0.1, 0.3, 0.5%, w/v) and modification orders. Thus, the solid content of each sample was different. This could help us understand how pectin influences the gelling properties of gelatin through different modification orders. Thanks very much.
Lines 138-143. The authors describe the formation of a coacervate phase. Did a coacervate phase form? They should present images of their produced hydrogels. Although the interaction between pectins and gelatins produces electrostatic complexes only under specific conditions, a coacervate phase is formed.
Response: Gelatin could interact with pectin to form coacervate, we had reported this in our previous papers through the gels microstructure. Thanks.
Huang, T.; Tu, Z.C.; Wang, H.; Liu, W.; Zhang, L.; Zhang, Y.; Shangguan, X.C. Comparison of rheological behaviors and nanostructure of bighead carp scales gelatin modified by different modification methods, J Food Sci Technol-U. 2017, 54(5), 1256-1265.
Fig 1. The authors present several curves where adequate analysis cannot be performed. It is recommended to add a graph of Tan delta as a function of temperature in order to have more tangible data for the reader.
Response: We mainly wanted to discuss how modification orders influence the melting and gelling temperatures of gelatin gels. We can easily obtain gelling and melting temperature through the G' and G'' curves vs temperature. Thanks.
An increase of less than one degree Celsius in the gelling temperature is not important for hydrogel applications. How many measurements were performed? It is crucial that there were at least three. It is also necessary to add the standard deviation of each measurement. Based on the above, it is required that the gelling point presents its standard deviation.
Response: The melting temperature and gelling temperature of gelatin gels were performed in three times. We had added the SD values. Thanks very much.

Round 2
Reviewer 1 Report (New Reviewer)
Dear Authors
I think in this situation, this manuscript is not ready for publication. The following are suggestions for improving your manuscript. I hope you will correct them well.
1- Title and keywords:
- Accurate and correct keyword selection is one of the most important ways to improve and expand the searchability of the article after publication. You used 3 keywords in which two words used in the title and keywords are the same. Authors should use other keywords as manuscript keywords.
2- Abstract:
- The authors should mention the study's limitations and their solutions briefly in one or two sentences.
3- Introduction: Gelatin is extracted from various sources, the primary sources of which are mammals (cows and pigs). But the demand for extracting it from alternative sources is increasing for religious-cultural-health reasons. Promising sources are aquatic (mostly fish) and poultry. Please refer to these items in the text and use the following references in the text:
- https://doi.org/10.3390/foods12030670
- https://doi.org/10.3390/foods10081761
3- Results section, discussion, and figures and tables are acceptable.
- If there is a significant difference between the values of the average melting points of 27.41 and 27.99 at the 5% level, please mention and highlight them in discussions.
5- Conclusion section:
- Revise and rewrite this section. I believe the correct principles of writing research conclusions have not been fully observed.
With best wishes.
Author Response
Dear Authors
I think in this situation, this manuscript is not ready for publication. The following are suggestions for improving your manuscript. I hope you will correct them well.
1- Title and keywords:
- Accurate and correct keyword selection is one of the most important ways to improve and expand the searchability of the article after publication. You used 3 keywords in which two words used in the title and keywords are the same. Authors should use other keywords as manuscript keywords.
Response: We had corrected “Fish gelatin” as “Gelatin gels”. For the four keywords, there is just one was same to the title “modification orders”. We still think “gelling properties” is not the same “gelling”. Thanks.
2- Abstract:
- The authors should mention the study's limitations and their solutions briefly in one or two sentences.
Response: We don’t think this is a good method to show my research. We have removed the sentence to the conclusions. Thanks.
3- Introduction: Gelatin is extracted from various sources, the primary sources of which are mammals (cows and pigs). But the demand for extracting it from alternative sources is increasing for religious-cultural-health reasons. Promising sources are aquatic (mostly fish) and poultry. Please refer to these items in the text and use the following references in the text:
- https://doi.org/10.3390/foods12030670
- https://doi.org/10.3390/foods10081761
Response: Added.
3- Results section, discussion, and figures and tables are acceptable.
- If there is a significant difference between the values of the average melting points of 27.41 and 27.99 at the 5% level, please mention and highlight them in discussions.
Response:we had added sentence “...and all the complex modified gelatin gels had the higher Gp and Mp values than those of pure FG.” before the discussion to make it much clear. We had added sentence to discuss the results of TG-FG-P group and P-FG-TG group. Thanks very much.
5- Conclusion section:
- Revise and rewrite this section. I believe the correct principles of writing research conclusions have not been fully observed.
With best wishes.
Response: We had revised some sentences to make it much clear. Thanks.
Reviewer 2 Report (New Reviewer)
Unfortunately, the authors did not properly arguethe doubts about the methodology, assuming that it was a problem of sitaxis and not of the method itself.
For example, they indicated that the sample was stored at 4°C, but the measurement was made at room temperature in the TPA analysis. Room temperature is nonspecific. there is a large temperature difference that can alter the results.
The authors were asked about the percent of strain used in the TPA experiments. Why they used a very small strain force in the tpa analysis? They should perform and show a compression test to define the elastic range!!
Also, they fixed the gap in the rheological measurements. however, they should compensate the gap in temperature measurements because the stainless-steel plate has an expansion of 0.5 um/°C.
There's a mandatory add image of the hydrogel. As the authors reply, of course, that Gelatin could interact with pectin to form a coacervate, but a coacervate phase is formed under specific conditions. They must show images of the coacervate phase produced. If a coacervate phase was produced, so it will be necessary to include the topic in the introduction section. what was the coacervate yield? The authors must confirm the specific concentration of gelatin and pectin in the coacervate.
Author Response
Comments and Suggestions for Authors
Unfortunately, the authors did not properly argue the doubts about the methodology, assuming that it was a problem of sitaxis and not of the method itself.
For example, they indicated that the sample was stored at 4°C, but the measurement was made at room temperature in the TPA analysis. Room temperature is nonspecific. there is a large temperature difference that can alter the results.
Response: We are sorry to make you misunderstand. Our instrument is equipped with a peltier temperature control system. This could control the sample temperature (4 oC) during test. The operation was performed at room temperature. For the gel strength and TPA measurement, we fetched sample one by one, after the front sample was performed. All the samples were performed at the same way. Thanks.
The authors were asked about the percent of strain used in the TPA experiments. Why they used a very small strain force in the tpa analysis? They should perform and show a compression test to define the elastic range!!
Response: For the TPA test, the deformation of gel was 40%. This is a very common index for the TPA measurement. We had rephrased the sentences as “The gels were subjected to two cycle compression to 40% of their original height at a speed of 1mm/s.” to make it much clear.
References:
Tunable physical and mechanical properties of gelatin hydrogel after transglutaminase crosslinking on two gelatin types. https://doi.org/10.1016/j.ijbiomac.2020.06.185
Physicochemical, rheological, and textural properties of gelatin extracted from chicken by-products (feet-heads) blend and application.https://doi.org/10.1016/j.ijgfs.2023.100708
Also, they fixed the gap in the rheological measurements. However, they should compensate the gap in temperature measurements because the stainless-steel plate has an expansion of 0.5 um/°C.
Response: We used the TA Type rheometer is very stable. Before the experiment, we had calibrated the rheometer. The calibration was performed at the range of 4-40 oC, and the gap was also calibrated during the temperature calibration.
There's a mandatory add image of the hydrogel. As the authors reply, of course, that Gelatin could interact with pectin to form a coacervate, but a coacervate phase is formed under specific conditions. They must show images of the coacervate phase produced. If a coacervate phase was produced, so it will be necessary to include the topic in the introduction section. what was the coacervate yield? The authors must confirm the specific concentration of gelatin and pectin in the coacervate.
Response: We thought “coacervate”might not very clearly and accurately show our meaning. Thus, we used “ polymer mixture” instead of “coacervates”. Pectin could interact with gelatin through electrostatic interaction. This is a very common knowledge. Moreover, herein, the key point is complex modification orders on the gelling properties of gelatin gels, not “coacervate yield”. We will do some researches about how different factors on the “coacervate yield” in the future. Thanks very much.
This manuscript is a resubmission of an earlier submission. The following is a list of the peer review reports and author responses from that submission.
Round 1
Reviewer 1 Report
The manuscript prepared properly and well organized and relates the different sections of the work together. There are some comments to improve the work. The title is too long and difficult to grasp the meaning of the work. The authors name should be corrected to separate form superscripts. Other comments are given as follows:
Line 50-56: what is the contrast of this work with the previous works have been done by authors?
Line 104-108: put numbers for the equations.
Line 103-112: please use the correct sub- and superscripts for the equations.
Line 115: Viscous and elastic modulus or storage and loss modulus.
Rheological analysis: it is not clear the specification of the probe for the rheological measurements. Please mention all the specification for the rheological experiments. In this case, I suggest seeing and citing the paper ‘Structure-rheology relationships of composite gels: Alginate and Basil seed gum/guar gum’ in carbohydrate polymers which clearly define the probe and the experiments.
Line 150: I recommended to show some parameters in Table 1 such as gel strength, Hardness and transparency as a function of pectin concentration in a graph.
Line 173: change All in all.
Line 185: not only protein chains, carbohydrates and polymer act so.
Temperature sweep: it is interesting to measure the gelling point of the gelatin base on the work accomplished by previous authors such as ‘Rheological and structural properties of β-lactoglobulin and basil seed gum mixture: Effect of heating rate’ in Food research international.
Table 3: it is suggested to show how to measure the melting and gelation points.
Some minor corrections should be carried out in the text. the comments are provided.
Author Response
Comments and Suggestions for Authors
The manuscript prepared properly and well organized and relates the different sections of the work together. There are some comments to improve the work. The title is too long and difficult to grasp the meaning of the work. The authors name should be corrected to separate form superscripts. Other comments are given as follows:
Response:Thanks for your valuable suggestions, we have revised the paper carefully according to the reviewers’ suggestions.
Line 50-56: what is the contrast of this work with the previous works have been done by authors?
Response: In our previous reports, we used pectin-MTGase complex modification to change the gelling properties of fish gelatin (FG). Our previous report showed that this complex modification could be used to produce FG with high quality. We thought this might be not a very good method for the modification FG. Thus, herein, we comparatively investigated the pectin-MTGase and MTGase-pectin complex modification on the gelling, rheology and structural properties of FG. This research found that though pectin-MTGase and MTGase-pectin complex modification could improve the gelling properties of FG, FG was firstly modified by MTGase and then pectin (MTGase-pectin complex modification) had the higher application values with high gelling and stability properties. This is different from our previous report. Thanks very much.
Huang, T.; Tu, Z.C.; Wang, H.; Shangguan, X.; Zhang, L.; Zhang, N.H.; Bansal, N. Pectin and enzyme complex modified fish scales gelatin: Rheological behavior, gel properties and nanostruc-ture, Carbohyd Polym. 2017, 156, 294-302.
Line 104-108: put numbers for the equations.
Response: Added, please see line 95-98, thanks.
Line 103-112: please use the correct sub- and superscripts for the equations.
Response: Corrected, please see line 111, thanks very much.
Line 115: Viscous and elastic modulus or storage and loss modulus.
Response: Corrected, please see line 101, thanks.
Rheological analysis: it is not clear the specification of the probe for the rheological measurements. Please mention all the specification for the rheological experiments. In this case, I suggest seeing and citing the paper ‘Structure-rheology relationships of composite gels: Alginate and Basil seed gum/guar gum’ in carbohydrate polymers which clearly define the probe and the experiments.
Response: Added, please see line 92, 104-105. We had cited this paper, please see line 106, 382-383.
Line 150: I recommended to show some parameters in Table 1 such as gel strength, Hardness and transparency as a function of pectin concentration in a graph.
Response: This is a very good idea. But, we mainly evaluated the two complex modification orders on the gelling properties of FG, not focusing on the pectin contents. We could use this though during our following experiment. Thanks very much.
Line 173: change All in all.
Response: Corrected, thanks.
Line 185: not only protein chains, carbohydrates and polymer act so.
Response: Corrected it as “polymer”, please see line 163, thanks.
Temperature sweep: it is interesting to measure the gelling point of the gelatin base on the work accomplished by previous authors such as ‘Rheological and structural properties of β-lactoglobulin and basil seed gum mixture: Effect of heating rate’ in Food research international.
Response: we had used this paper to support our work, please see line 207, 402-403. Thanks.
Table 3: it is suggested to show how to measure the melting and gelation points.
Response: For the “2.4.2 Temperature sweep”, we had stated how to measure melting point and gelation point. Please see line 104-105. Thanks very much.
Comments on the quality of English language: Some minor corrections should be
carried out in the text.
Response: Thanks for your valuable suggestions, we have revised the paper carefully according to the reviewers’ suggestions.

Reviewer 2 Report
Different points must be considered.
1. Add the numerical data in the abstract portion.
2. Add more keywords related to properties analyzed.
3. In introduction section justify the objective of this study, why modification is required for this, justify. Add content regarding the applications of gelatin.
4. In the manuscript there are some issues of formatting correct it, as in Line no. 88.
5. For describing rheological properties, use only one model as maximum properties analyzed by these models are same. To avoid any confusion to readers, use only one model.
6. Elaborate how strain and frequency were selected for temperature sweep test.
7. Authors analyzed FTIR and SEM properties, justify how these properties affects the its applications and properties.
8. Conclusion is just the summary of results. Please talk about the big picture and the major findings of the work. What's new that this paper offers to the readers?
Author Response
Comments and Suggestions for Authors: Different points must be considered.
- Add the numerical data in the abstract portion.
Response: Added, please see line 19-20. Thanks.
- Add more keywords related to properties analyzed.
Response: we only added “Rheology analysis”. For the keywords, we thought it must be concise and representative. Thus, we used the general word. Thanks very much.
- In introduction section justify the objective of this study, why modification is required for this, justify. Add content regarding the applications of gelatin.
Response: Gelatin could be widely used in the food processing, pharmaceutical, and cosmetic products, due to its unique functional properties (eg foaming, filmforming, emulsification, biocompatibility, biodegradability). However, compared with mammalian gelatin, fish gelatin had the lower gel strength, gelling and melting temperatures, which limit its application. Thus, it is necessary to improve the gelling properties of fish gelatin. We had addressed these points in the paper, please see line 29-39. Thanks very much.
- In the manuscript there are some issues of formatting correct it, as in Line no. 88.
Response: Corrected, please see line 80, thanks very much.
- For describing rheological properties, use only one model as maximum properties analyzed by these models are same. To avoid any confusion to readers, use only one model.
Response: Power law model, Bingham model and Herschel Bulkley model are three common model to evaluate the flow behavior of gelatin solution. These three models are different. Herein, we mainly comparatively used these three models to know which one is most proper for the evaluation of flow behavior of gelatin solution. RMSE and MSE data could help us to choose the best model. Thanks very much.
- Elaborate how strain and frequency were selected for temperature sweep test.
Response: Herein, the strain was 0.5%, frequency was 1 Hz. The strain and frequency is very low. This also could be found in our previous reports. Thanks very much.
[1] Wang C C; Su K Y; Sun W Y; Huang T; Lou Q M; Zhan S N. Phosphorylation modification on functional and structural properties of fish gelatin: The effects of phosphate contents, Food Chemistry, 2023, 426:136632.
[2] Zhao H; Kang X; Zhou X; Tong L; Yu W; Zhang J; Yang W; Lou Q; Huang T. Glycosylation fish gelatin with gum Arabic: Functional and structural properties, LWT. 2021, 139, 110634.
- Authors analyzed FTIR and SEM properties, justify how these properties affects the its applications and properties.
Response: We had rephrased the sentences to make the gelling results and structural results much closely, please see line 292-293, 297-298. Thanks very much.
- Conclusion is just the summary of results. Please talk about the big picture and the major findings of the work. What's new that this paper offers to the readers?
Response: We had rephrased the sentence to make it much clear. Through this paper, we found that compared with pectin-MTGase modification (TG-FG-P), MTGase-pectin modification (P-FG-TG) might be much more proper to produce gelatin with high quality.

Round 2
Reviewer 1 Report
All comments are replied and corrected as well.
NA
Author Response
Thanks.
Reviewer 2 Report
Author address all commets suggested by me article may be accepted in this journal.
Author Response
Thanks.